# Effects of Phosphorus Supply on Seed Yield and Quality in Flax

Yaping Xie [1,†], Yang Li [2,†], Yanni Qi [1], Limin Wang [1], Wei Zhao [1], Wenjuan Li [1], Zhao Dang [1], Jianping Zhang [1,*], Xingzhen Wang [1], Yanjun Zhang [1], Xingrong Wang [1], Zhengjun Cui [3], Linrong Shi [4] and Zhongcheng Lv [5]

1   Crop Research Institute, Gansu Academy of Agricultural Sciences (GASS), Lanzhou 730070, China
2   College of Agronomy, Shanxi Agricultural University (SXAU), Taiyuan 030006, China
3   College of Agronomy, Gansu Agricultural University (GSAU), Lanzhou 730070, China
4   College of Mechanical and Electrical Engineering, Gansu Agricultural University, Lanzhou 730070, China
5   Ordos Institute of Agricultural Sciences (OIAS), Dongsheng 017000, China
*   Correspondence: zhangjpzw3@gsagr.ac.cn; Tel.: +86-0931-761-108-1
†   These authors contributed equally to this work.

**Abstract:** Flaxseed, which is rich in lignan, α-linolenic acid, dietary fiber, and several minerals, is an important food and nutrition source. In this study, trials were conducted at Yongdeng over two consecutive years (2018 and 2019), with three cultivars (Longyaza 1, Longya 14, and Zhangya 2) and five application rates of phosphorus (P) (0, 40, 80, 120, and 160 kg $P_2O_5$ ha$^{-1}$). We examined the effects of P on the seed yield, and the yields and contents of dietary fiber, lignan, iron (Fe), zinc (Zn), manganese (Mn), and copper (Cu). We found that P fertilization positively influenced yields of seed and levels of lignan, Fe, and Cu, showing average increases of 15, 20, 24, and 28%, respectively, compared with plants not given P over the 2-year study. Additionally, P fertilization resulted in increased concentrations of Fe and Cu in flaxseed of 8 and 2%, respectively. P fertilization negatively affected the levels of dietary fiber, Zn, and Mn, which were reduced by 7, 11, and 7%, respectively, in comparison with the control. In conclusion, the results demonstrated that appropriate P application is an effective strategy for improving yields of seed, lignan, Fe, and Cu in flax production and for enhancing concentrations of Fe and Cu in flax.

**Keywords:** phosphorus; flaxseed; lignan; iron; copper; zinc

## 1. Introduction

Flax (*Linum usitatissimum* L.) is an important oil crop which is cultivated widely across the world, and its seed is an important source of food and nutrition. In China, with ongoing improvements in people's living standards, flax is increasingly becoming appreciated by consumers and researchers as it contains abundant biologically active components such as omega-3 fatty acids, lignan, dietary fiber, and minerals [1–4]. Population growth and the decline in the area of available arable land have caused maximizing crop yield per unit area to become a major aim of agricultural production in China [5].

Globally, many developing countries are facing mineral nutrient deficiencies in their populations [6]. More than two billion people globally are affected by deficiencies of key micronutrients, including Fe and Zn [7]. Flaxseed is a good source of minerals, especially Fe and Zn [2]. The biofortification of crops through application of fertilizers is an effective approach to regulate the contents of mineral elements in crops [6]. Thus, application of phosphorus (P) fertilization is a commonly used agronomic practice to optimize seed yield and crop composition. Nowadays, much attention is focused on the regulation of flaxseed components for human nutrition.

Numerous studies have shown that the application of an appropriate quantity and intensity of P can notably increase the seed yields of flax [8–10], soybean [11], canola [11,12], wheat [13,14], maize [14,15], and rice [14]. In addition, many studies have documented the effect of P on concentrations of micronutrient minerals in crop seeds. A study on flaxseed

revealed lower Zn concentrations at higher P rates [16]. Similar results have been reported for other plant species including wheat [13,14,17], maize [15], and rice [14], where increased P fertilizers also resulted in reduced grain Zn concentrations. Esmail et al. [8] found greater Fe concentrations in flaxseed with increased P fertilizer rates under dryland conditions. In a greenhouse study conducted in Ottawa, concentrations of Zn, Mn, and Fe in canola seed decreased with an increasing P supply, while the P supply level had no obvious effects on the Cu concentration [11]. However, another study on wheat showed that P fertilizer had a negligible effect on the concentration of Fe in grain [17]. In addition, another experiment on wheat indicated that levels of Fe, Mn, and Cu in grain either remained the same or increased with the application of P [13]. In maize, Zhang et al. (2017) [18] found that P supply decreased Zn and Cu concentrations in the grain. Similarly, Zhang et al. (2020) [19] found that P application significantly decreased the average Cu concentration but had no significant effect on the Fe concentration in the grain of maize.

The simultaneous improvement of yield and quality has become an important goal for modern flax production in China. Therefore, the aim of this study was to explore the effects of the P supply rate on the seed yield and the yields and concentrations of dietary fiber, lignan, and trace elements including Fe, Zn, Mn, and Cu, in flaxseed.

## 2. Materials and Methods

### 2.1. Experimental Site

The effects of the P application rate on the dietary fiber, protein, and mineral contents in flaxseed were investigated in a field experiment conducted at Yongdeng County of Gansu Academy of Agricultural Sciences Experimental Station, Gansu, China (36°02′ N, 103°40′ E, altitude 2149 m) from 2018 to 2019. Data on the air temperature and precipitation were recorded daily during the 2018 and 2019 growing seasons and are shown in Figure 1. Overall, regarding precipitation during the growing season, in 2018 there was appreciable rainfall with greater rainfall frequency, and the weather cleared up quickly in July when the flax was in the middle of the seed-filling period. In 2019 there was concentrated rainfall in August during the later seed-filling period. In the present study, the seed yield in 2018 was greater than that of 2019. This was due to a higher amount of rainfall in 2018 in the middle of the seed-filling period due to the greater frequency of rainfall, then the weather cleared quickly. Under these conditions, the greater seed yield in 2018 was perhaps due to the sunshine duration, illumination intensity, humidity, soil moisture, soil microorganisms, etc., being more favorable for seed-filling. However, in 2019, due to the concentrated rainfall, serious conflicts between water supply and crop demand led to lodging and disease (Figure 1).

The soil type was arenosol [20], with wheat as the previous crop for two years. Prior to the application of P fertilizer, soil samples were collected from a depth of 0–30 cm. The soil samples were air dried, ground in a mortar, and passed through a 1 mm plastic sieve. The chemical characteristics of the soil samples (from 0–30 cm) were analyzed using the methods reported by Lithourgidis et al. [21]. Levels of available P in the soil were determined using the colorimetric molybdenum-blue method [21]. The baseline of available P was 8.0 mg kg$^{-1}$ at a soil depth of 0–30 cm, considerably lower than the optimum P of above 20 mg kg$^{-1}$ [22]. Zinc, Fe, Mn, and Cu in the soil were extracted with 5 m mol DTPA (diethylene triamine pentacetic acid) at pH 7.3 (180 rpm, 25 °C) and analyzed with an inductively coupled plasma optical emission spectrometer (ICP-MS, Agilent 7900, Palo Alto, California, USA) [23]. The concentrations of nutrients and mineral elements in the soil are listed in Table 1.

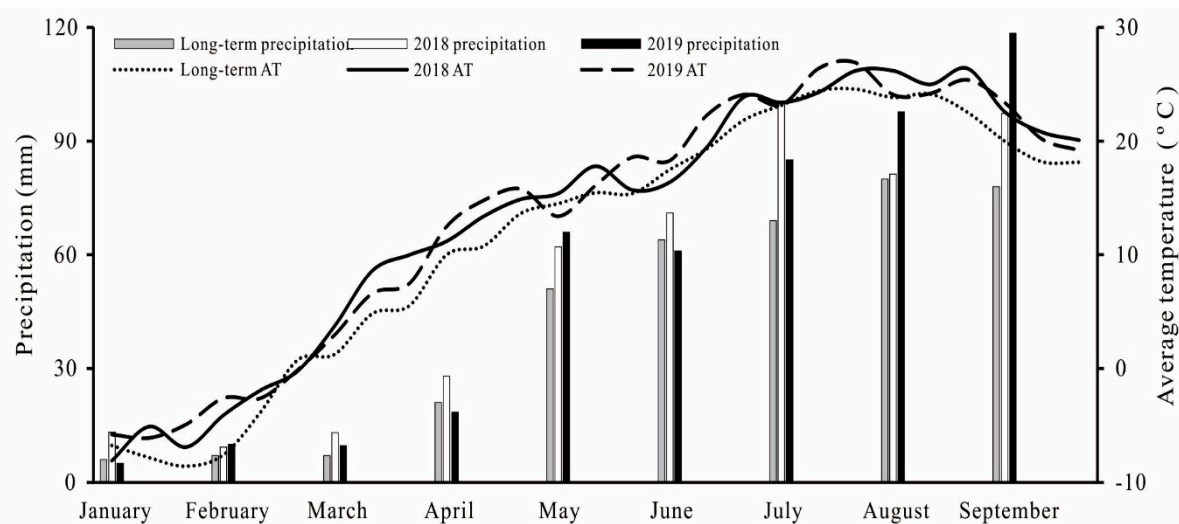

**Figure 1.** Monthly mean air temperatures and precipitation for 2018, 2019, and the 30-year average at Yongdeng, China.

**Table 1.** General soil properties and DTPA-extractable mineral elements before sowing in 2018 and 2019.

| Year | Organic Matter | Alkali-Hydrolyzable Nitrogen | Available P [a] | Available Potassium | pH | DTPA-Fe | DTPA-Zn | DTPA-Mn | DTPA-Cu |
|------|------|------|------|------|------|------|------|------|------|
| | g kg$^{-1}$ | mg kg$^{-1}$ | | | | mg kg$^{-1}$ | | | |
| 2018 | 9.8 | 53.9 | 8.0 | 178.3 | 7.5 | 3.95 | 067 | 4.37 | 1.50 |
| 2019 | 7.6 | 48.2 | 8.7 | 141.6 | 8.2 | 3.12 | 0.54 | 4.01 | 1.20 |

[a] P: phosphorus; DTPA-Fe, DTPA-Zn, DTPA-Mn, and DTPA-Cu: diethylene triamine pentacetic acid extraction of iron (Fe), zinc (Zn), manganese (Mn), and copper (Cu), respectively.

On 2 August 2018 and 3 August 2019, the harvest dates, the crops in each plot were separately harvested using a sickle to determine the seed yield.

### 2.2. Experimental Design

Randomized complete block design with three replicates was used for this study, with a plot size of 20 m$^2$ (4 m × 5 m). Five P rates (0, 40, 80, 120, and 160 kg P$_2$O$_5$ ha$^{-1}$) (defined as P$_0$ (also serving as a control), P$_{40}$, P$_{80}$, P$_{120}$, and P$_{160}$, respectively) were applied to three flax cultivars (Longyaza 1, Longya 14, and Zhangya 2). Sowing density was at a rate of 1050 viable seeds m$^{-2}$. Seeds were sowed using an in-house constructed planter and the sowing depth was 3 cm. Phosphorus fertilizer was applied pre-planting in the form of calcium superphosphate broadcast uniformly over the soil surface prior to seedbed preparation and sowing. Nitrogen (N) at 60 kg N ha$^{-1}$ and potassium (K) at 52.5 kg K$_2$O ha$^{-1}$ were each applied using urea and potassium sulfate, respectively, each year at pre-planting before seedbed preparation and sowing. An additional 20 kg N ha$^{-1}$ of urea was top-dressed at the budding stage of the flax. The flax was kept free of weeds by hand hoeing. To ensure maximum potential productivity, all plots received 40 mm irrigation prior to the flax flowering, using pipes of 13 cm diameter. A water meter installed at the discharging end of the pipes measured and recorded the amount of irrigation. Further crop management procedures followed common agricultural practices.

### 2.3. Lignan and Dietary Fiber Concentration Determination

The lignan concentration was determined using a near-infrared reflectance diode array analyzer (Perten Instruments, Stockholm, Sweden) according to a previously reported method [24]. The calibration was carried out by Thermo Galactic Grams PLS IQ software (Perten Instruments, Stockholm, Sweden). The calibration curve was updated yearly on the

basis of independent samples analyzed by high-performance liquid chromatography [9]. The dietary fiber content of the seeds was quantified using an enzymatic gravimetric method [25].

### 2.4. Determination of Fe, Zn, Mn, and Cu Concentrations

The seeds from each plot were ground separately using a mill to pass through a 1 mm screen. The concentrations of Fe, Zn, Mn, and Cu were determined using an inductively coupled plasma mass spectrometer (Agilent 7900, Agilent Technologies, Palo Alto, California, USA). Details of the procedures were as described by Singh et al. [26].

### 2.5. Lignan, Dietary Fiber, Fe, Zn, Mn, and Cu Yields

Lignan, dietary fiber, Fe, Zn, Mn, and Cu yields were calculated according to the following formulas:

$$\text{Lignan yield (g ha}^{-1}) = \text{lignan concentration (g kg}^{-1}) \times \text{seed yield (kg ha}^{-1}) \quad (1)$$

$$\text{Dietary fiber yield (kg ha}^{-1}) = \text{Dietary fiber content (g kg}^{-1}) \times \text{seed yield (kg ha}^{-1}) \quad (2)$$

$$\text{Fe (Zn, Mn, and Cu) yield (g ha}^{-1}) = \text{Fe (Zn, Mn, and Cu) concentration (mg kg}^{-1}) \times \text{seed yield (kg ha}^{-1}) \quad (3)$$

### 2.6. Statistical Analysis

Data were analyzed using the SPSS 22.0 software package (IBM Corp., Chicago, IL, USA). The Tukey test ($p = 0.05$) was performed to identify significant differences among means. Regression analyses were computed to evaluate the relationships between the P rates and the variables.

## 3. Results

### 3.1. Seed Yield

Seed yield was affected by the year, P level, cultivar, and the interaction of year and cultivar (Table 2). On average, there were greater seed yields in 2018 than in 2019. It was noted that P supply significantly influenced the seed yield of flax. Seed yield increased as P supply increased up to 120 kg $P_2O_5$ ha$^{-1}$ (averaging 1751 kg ha$^{-1}$), then declined (Figure 2A). Phosphorus application significantly improved the seed yield by an average of 219 kg ha$^{-1}$ compared with no P treatment, over three cultivars and both years. Among cultivars, Longya 14 obtained the highest yield (Figure 2B).

**Table 2.** Analysis of variance results for the dependent variables.

| Dependent Variable | Y [a] | P | C | Y × P | Y × C | P × C | Y × P × C |
|---|---|---|---|---|---|---|---|
| | | | | *F*-value | | | |
| Seed yield | 51.21 * | 24.58 ** | 23.51 * | 1.40 | 5.19 * | 2.72 | 1.36 |
| Lignan concentration | 284.19 ** | 1.28 | 150.28 ** | 1.64 | 0.86 | 0.70 | 0.39 |
| Lignan yield | 52.21 * | 10.11 * | 18.70 * | 1.98 | 11.13 ** | 1.74 | 1.17 |
| Dietary fiber concentration | 1040.64 * | 9.66 * | 28.27 * | 1.61 | 0.24 | 0.87 | 2.54 * |
| Dietary fiber yield | 133.71 * | 3.18 | 175.57 ** | 1.64 | 0.61 | 2.98 | 0.94 |
| Iron concentration | 36.31 * | 11.53 * | 4.74 | 1.52 | 2.78 | 1.11 | 3.88 ** |
| Iron yield | 42.18 * | 167.46 ** | 8.79 * | 0.14 | 5.91 * | 0.68 | 4.92 ** |
| Zinc concentration | 1.34 | 13.28 * | 22.39 * | 2.76 | 0.59 | 0.73 | 3.98 ** |
| Zinc yield | 44.70 * | 2.16 | 30.69 * | 1.41 | 2.01 | 1.22 | 3.30 ** |
| Manganese concentration | 62.17 * | 15.04 * | 2.59 | 0.27 | 3.50 | 1.06 | 8.34 ** |
| Manganese yield | 44.03 * | 2.75 | 0.31 | 0.60 | 7.27 * | 1.52 | 5.83 ** |
| Copper concentration | 13.91 * | 63.53 ** | 1.79 | 0.36 | 12.99 ** | 3.13 | 0.48 |
| Copper yield | 44.93 * | 34.59 ** | 3.02 | 2.01 | 13.67 ** | 5.39 * | 0.42 |

[a] Y: year; P: phosphorus level; C: cultivar. * Significant at the 0.05 probability level. ** Significant at the 0.01 probability level.

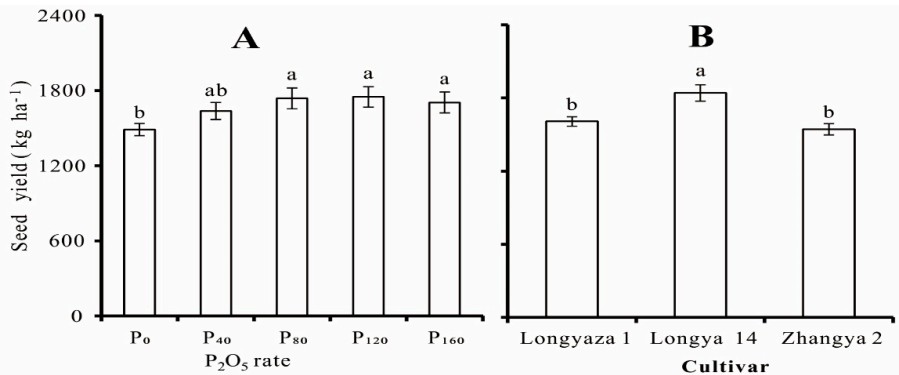

**Figure 2.** Effects of the P fertilization level (A) and cultivar (B) on the seed yields of flax across 2018 and 2019 in Yongdeng, China. Data are expressed as means $\pm$ standard errors (SE) ($n = 15$). Different letters indicate means that are significantly different at $p < 0.05$ according to the Tukey test.

### 3.2. Lignan Concentration and Yield

The year and cultivar had significant impact on the lignan concentration (Table 2). The lignan concentration was 2.0 g kg$^{-1}$ greater in 2018 than in 2019. Different cultivars showed considerable differences in lignan concentration. Longya 14 had the highest lignan level of 9.1 g kg$^{-1}$, while Zhangya 2 showed the lowest level at 7.2 g kg$^{-1}$ (Figure 3A).

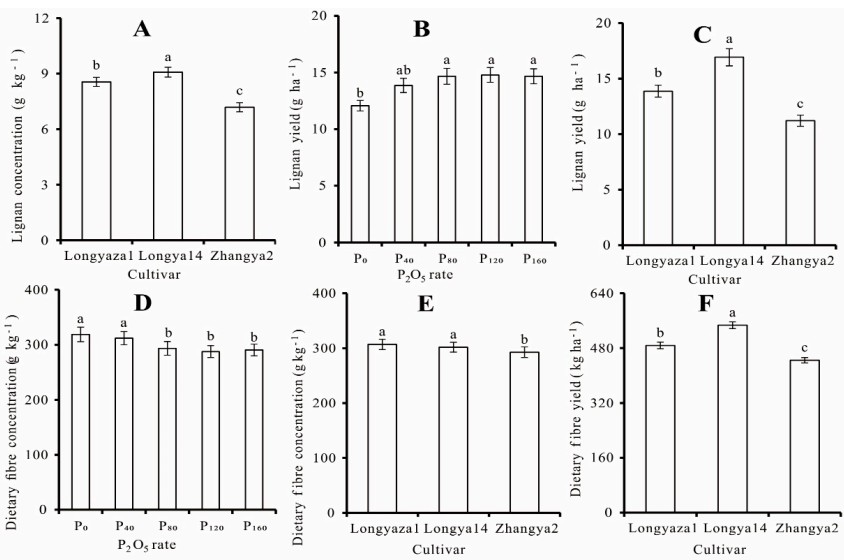

**Figure 3.** Effects of (**A**) cultivar on lignan concentration, (**B**) phosphorus level on lignan yield, (**C**) cultivar on lignan yield, (**D**) phosphorus level on dietary fiber concentration, and (**E**) cultivar on dietary fiber concentration and (**F**) yield. Error bars are standard errors of the means ($n = 15$). Different letters indicate means that are significantly different at $P < 0.05$ according to the Tukey test.

Meanwhile, the year, P supply, and cultivar impacted the lignan yield (Table 2). The increase in lignan yield was 5.7 g ha$^{-1}$ in 2018 compared with 2019 (Table 3). The lignan yield increased from 0 to 80 kg P$_2$O$_5$ ha$^{-1}$ and then plateaued between 80 and 160 kg P$_2$O$_5$ ha$^{-1}$ (Figure 3B). The change trend of lignan yield as a function of the P level was consistent with that of the seed yield. On average, P$_{40}$, P$_{80}$, P$_{120}$, and P$_{160}$ supported a greater lignan yield (14.5 g ha$^{-1}$) than P$_0$ (12.1 g ha$^{-1}$). Longya 14 showed the maximal lignan yield (Figure 3C). Furthermore, the interaction of year and cultivar affected the lignan yield; Longya 14 obtained the highest lignan yield in 2018 (20.6 g ha$^{-1}$), and Zhangya 2 exhibited the lowest lignan yield in 2019 (8.7 g ha$^{-1}$) (Table 4).

**Table 3.** Lignan and dietary fiber concentrations and yields for three cultivars and five phosphorus rates.

| Treatments | Lignan Concentration | | Lignan Yield | | Dietary Fiber Concentration | | Dietary Fiber Yield | |
|---|---|---|---|---|---|---|---|---|
| | **2018** | **2019** | **2018** | **2019** | **2018** | **2019** | **2018** | **2019** |
| | g kg$^{-1}$ | | g ha$^{-1}$ | | g kg$^{-1}$ | | kg ha$^{-1}$ | |
| $P_0$ | 9.01 | 7.01 | 14.45 b | 9.69 b | 265.3 a | 372.4 a | 425.09 | 512.97 |
| $P_{40}$ | 9.15 | 7.09 | 16.75 b | 10.95 b | 264.9 ab | 359.1 a | 458.80 | 545.34 |
| $P_{80}$ | 9.40 | 7.18 | 18.03 a | 11.29 a | 244.0 c | 343.0 b | 464.57 | 546.61 |
| $P_{120}$ | 9.46 | 7.46 | 17.52 a | 12.04 a | 244.2 c | 330.9 b | 465.46 | 529.16 |
| $P_{160}$ | 9.48 | 7.49 | 17.65 a | 11.68 a | 247.9 bc | 333.2 b | 465.77 | 515.03 |
| Lonyaza 1 | 9.54 b [†] | 7.58 a | 16.38 b | 11.36 b | 260.4 a | 341.5 | 446.42 b | 528.79 b |
| Longya 14 | 10.20 a | 7.96 a | 20.56 a | 13.29 a | 255.4 ab | 348.2 | 512.84 a | 580.11 a |
| Zhangya 2 | 8.16 c | 6.20 b | 13.69 c | 8.74 c | 243.9 b | 353.5 | 408.55 c | 480.57 c |

[†] Means in the same column followed by the same letter do not differ significantly according to the Tukey test ($p = 0.05$). The absence of letters in the same column symbolizes that there are no statistically significant differences according to the Tukey test ($p = 0.05$).

**Table 4.** Interaction of year, phosphorus levels, and cultivar relating to dietary fiber concentration, and concentrations and yields of Fe, Zn, and Mn.

| Treatment | Dietary Fiber c [a] | Iron (Fe) c | Fe Yield | Zinc (Zn) c | Zn Yield | Manganese (Mn) c | Mn Yield |
|---|---|---|---|---|---|---|---|
| Longyaza 1-2018-$P_0$ | 274.10 ef[†] | 78.93 defg | 126.43 efg | 44.51 bcd | 71.31 def | 66.27 a | 56.15 g |
| Longyaza 1-2018-$P_{40}$ | 277.87 e | 79.50 cdef | 135.79 def | 43.24 cdef | 73.85 de | 60.23 abcd | 57.40 g |
| Longyaza 1-2018-$P_{80}$ | 249.37 efg | 80.27 cdef | 140.61 cde | 40.03 fghi | 70.10 defg | 62.44 abc | 57.55 g |
| Longyaza 1-2018-$P_{120}$ | 253.03 efg | 82.80 bcde | 147.37 cd | 39.40 hijk | 70.07 defg | 50.90 fghi | 58.62 g |
| Longyaza 1-2018-$P_{160}$ | 247.60 efg | 84.47 abcd | 146.99 cd | 40.32 fghi | 70.16 defg | 58.69 bcde | 60.53 g |
| Longya 14-2018-$P_0$ | 270.60 ef | 86.63 abc | 145.60 cd | 46.38 abc | 77.92 cd | 56.39 cdef | 62.44 g |
| Longya 14-2018-$P_{40}$ | 261.67 efg | 86.56 abc | 168.39 b | 46.08 abc | 89.70 ab | 62.46 abc | 64.23 g |
| Longya 14-2018-$P_{80}$ | 236.40 g | 88.70 ab | 192.70 a | 44.23 bcde | 96.15 a | 60.88 abcd | 64.68 g |
| Longya 14-2018-$P_{120}$ | 246.50 gf | 91.00 a | 197.55 a | 42.73 defg | 92.75 ab | 55.20 defg | 65.14 g |
| Longya 14-2018-$P_{160}$ | 261.87 efg | 91.61 a | 192.42 a | 40.46 fghi | 84.92 bc | 56.27 cdef | 67.26 g |
| Zhangya 2-2018-$P_0$ | 251.17 efg | 77.67 efgh | 117.90 fghi | 46.25 abc | 70.21 defg | 65.25 ab | 67.36 g |
| Zhangya 2-2018-$P_{40}$ | 255.10 efg | 76.86 fghi | 124.62 efgh | 44.16 bcde | 71.60 def | 64.56 ab | 67.76 g |
| Zhangya 2-2018-$P_{80}$ | 246.13 gf | 86.52 l | 149.77 bcd | 40.60 fghi | 70.29 defg | 58.66 bcde | 68.50 g |
| Zhangya 2-2018-$P_{120}$ | 233.07 g | 86.14 abcd | 151.55 bcd | 39.56 ghijk | 69.61 defg | 62.69 abc | 69.66 g |
| Zhangya 2-2018-$P_{160}$ | 234.13 g | 88.50 ab | 155.85 bc | 40.65 fghi | 71.59 def | 53.02 efgh | 70.79 g |
| Longyaza 1-2019-$P_0$ | 380.00 a | 62.59 l | 85.56 m | 44.15 bcde | 60.53 ghi | 49.21 ghij | 90.60 f |
| Longyaza 1-2019-$P_{40}$ | 375.67 ab | 62.63 l | 91.70 klm | 41.07 efgh | 60.19 ghi | 44.12 ijkl | 93.37 f |
| Longyaza 1-2019-$P_{80}$ | 350.33 abcd | 68.62 jkl | 107.40 hijk | 40.30 fghi | 63.04 fgh | 43.46 jklm | 94.70 ef |
| Longyaza 1-2019-$P_{120}$ | 328.33 d | 78.37 efg | 122.52 efgh | 38.33 ijk | 59.89 ghi | 36.65 mno | 99.03 ef |
| Longyaza 1-2019-$P_{160}$ | 333.00 cd | 81.90 bcde | 125.55 efgh | 37.01 k | 56.71 hi | 39.50 lmno | 101.64 ef |
| Longya 14-2019-$P_0$ | 363.33 abc | 70.29 hijk | 101.45 jklm | 48.44 a | 69.91 defg | 38.88 lmno | 102.10 ef |
| Longya 14-2019-$P_{40}$ | 355.67 abcd | 73.92 ghijk | 124.72 efgh | 44.49 bcd | 75.05 cde | 41.27 klmn | 102.84 def |
| Longya 14-2019-$P_{80}$ | 349.00 bcd | 76.03 fghi | 132.76 def | 42.69 defg | 77.88 cd | 32.97 o | 104.60 cdef |
| Longya 14-2019-$P_{120}$ | 334.33 cd | 76.30 fghi | 133.22 def | 37.69 jk | 65.86 efgh | 35.77 no | 106.21 bcdef |
| Longya 14-2019-$P_{160}$ | 338.67 cd | 80.14 cdef | 137.74 cde | 40.62 fghi | 69.86 defg | 34.08 o | 109.36 bcde |
| Zhangya 2-2019-$P_0$ | 374.00 ab | 68.39 kl | 90.26 lm | 47.34 ab | 62.47 fgh | 48.63 ghij | 110.34 bcde |
| Zhangya 2-2019-$P_{40}$ | 346.00 bcd | 70.08 ijk | 98.09 jklm | 44.51 bcd | 62.31 ghi | 48.05 hijk | 118.13 abcd |
| Zhangya 2-2019-$P_{80}$ | 329.67 d | 77.19 efgh | 113.03 ghij | 44.57 bcd | 62.49 fgh | 48.48 ghij | 119.79 abc |
| Zhangya 2-2019-$P_{120}$ | 330.00 d | 80.11 cdef | 118.77 fghi | 38.23 ijk | 56.72 hi | 43.96 ijkl | 121.60 ab |
| Zhangya 2-2019-$P_{160}$ | 328.00 d | 80.08 cdef | 110.36 ghij | 37.55 jk | 51.79 i | 49.57 ghij | 132.40 a |

Note: [a] c: concentration;. [†] Means in the same column followed by the same letter do not differ significantly according to the Tukey test ($p = 0.05$).

### 3.3. Dietary Fiber Concentration and Yield

The dietary fiber concentrations and yields of flaxseed showed significant differences between the years (Tables 2 and 3). Specifically, the dietary fiber concentration and yield decreased by averages of 94.5 g kg$^{-1}$ and 73.9 kg ha$^{-1}$ in 2018, respectively, compared with 2019. Phosphorus application significantly affected the dietary fiber concentration but did not impact the dietary fiber yield (Table 2). In general, the dietary fiber concentration decreased by an average of 29.6 g kg$^{-1}$ with P application compared with $P_0$ (Figure 3D). Furthermore, the dietary fiber concentration and yield differed significantly among cultivars (Table 2). Longyaza 1 showed the highest dietary fiber concentration, while the maximal

dietary fiber yield was observed for Longya 14 (Figure 3E,F). Additionally, the effects of the interactions between year, P rate, and cultivar on dietary fiber concentration were observed. The highest concentration was in Longya 14 at $P_0$ in 2019 (380 g kg$^{-1}$), and Zhangya 2 had the lowest concentration at $P_{120}$ in 2018 (233.1 g kg$^{-1}$) (Table 4).

### 3.4. Iron Concentration and Yield

The Fe concentration and yield of flaxseed differed significantly between the examined years (Table 2). Averaging the results, the Fe level and yield in 2018 were 84.4 mg kg$^{-1}$ and 152.9 g ha$^{-1}$, respectively, notably higher than those in 2019 (Table 5). The application of P led to obvious increases of Fe concentration in the seed (Figure 4A). Nevertheless, no difference existed between $P_0$, $P_{40}$, and $P_{80}$, nor between $P_{80}$, $P_{120}$, and $P_{160}$.

**Table 5.** Iron and zinc concentrations and yields with three cultivars and five phosphorus levels.

| Treatments | Fe [a] Concentration | | Fe Yield | | Zn Concentration | | Zn Yield | |
|---|---|---|---|---|---|---|---|---|
| | 2018 | 2019 | 2018 | 2019 | 2018 | 2019 | 2018 | 2019 |
| | mg kg$^{-1}$ | | g ha$^{-1}$ | | mg kg$^{-1}$ | | g ha$^{-1}$ | |
| $P_0$ | 81.08 b [†] | 67.09 d | 129.98 b | 92.42 c | 45.71 a | 46.64 a | 73.14 b | 59.45 b |
| $P_{40}$ | 80.97 b | 68.88 cd | 142.93 ab | 104.84 bc | 44.49 a | 43.36 b | 75.56 b | 60.83 b |
| $P_{80}$ | 85.16 ab | 73.95 bc | 161.03 a | 117.73 ab | 41.62 b | 42.52 b | 77.48 a | 64.30 ab |
| $P_{120}$ | 86.64 a | 78.26 ab | 165.49 a | 124.83 a | 40.56 b | 38.08 c | 78.38 a | 65.85 ab |
| $P_{160}$ | 88.19 a | 80.71 a | 165.08 a | 124.55 a | 40.48 b | 38.39 c | 78.85 a | 67.80 a |
| Longyaza 1 | 81.19 b | 70.87 | 139.44 b | 106.55 b | 41.50 b | 40.17 | 71.10 b | 60.07 b |
| Longya 14 | 88.90 a | 75.17 | 179.33 a | 125.98 a | 43.98 a | 43.16 | 88.29 a | 71.71 a |
| Zhangya 2 | 83.14 b | 75.34 | 139.94 b | 106.10 b | 42.24 ab | 42.07 | 70.66 b | 59.16 b |

[a] Fe: iron; Zn: zinc. [†] Means in the same column followed by the same letter do not differ significantly according to the Tukey test ($p = 0.05$). The absence of letters in the same column indicates no statistically significant difference according to the Tukey test ($p = 0.05$).

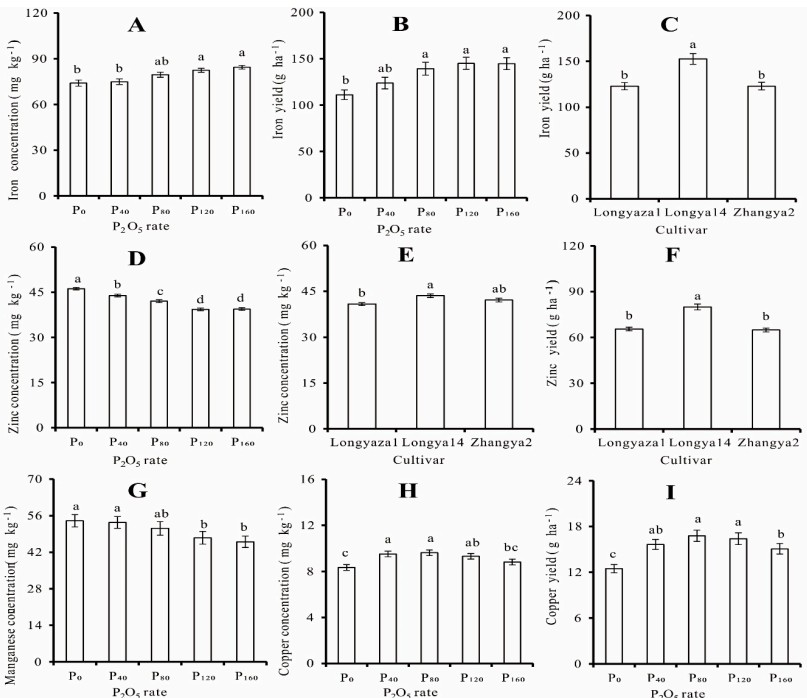

**Figure 4.** Effects of (**A**) phosphorus level and cultivar on iron concentration and (**B**) yield, (**C**) cultivar on iron yield, (**D**) phosphorus level on zinc concentration, (**E**) cultivar on zinc concentration and (**F**) yield, and phosphorus level on (**G**) manganese concentration, (**H**) copper concentration, and (**I**) copper yield. Error bars are standard errors of the means $\pm$ SE ($n = 15$). Different letters indicate means that are significantly different at $P < 0.05$ according to the Tukey test.

In the present study, we found that the Fe yield was affected by the P rate (Table 2). Specifically, the highest Fe yield was obtained at 120 kg $P_2O_5$ ha$^{-1}$ (Figure 4B). Among cultivars, Longya 14 obtained the most substantial Fe yield (Figure 4C). Moreover, the interaction year, P level, and cultivar influenced the Fe yield (Table 2). The greatest Fe yield was gained by Longya 14 at $P_{120}$ in 2018 (197.6 g ha$^{-1}$), and the lowest by Longyaza 1 at $P_0$ in 2019 (85.6 g ha$^{-1}$) (Table 4).

### 3.5. Zinc Concentration and Yield

The Zn concentration and yield of flaxseed were impacted by the interaction of year, P level, and cultivar (Table 2). The highest Zn concentration was achieved by Longya 14 sown at $P_0$ in 2019 (48.4 mg kg$^{-1}$), while the lowest value was found in Longyaza 1 at $P_{160}$ in 2019 (37.0 mg kg$^{-1}$) (Table 4). The highest Zn yield was observed for Longya 14 at $P_{80}$ in 2018 (96.2 g ha$^{-1}$), and the lowest yield was for Zhangya 2 at $P_{160}$ in 2019 (51.8 g ha$^{-1}$) (Table 4).

There were significant differences in Zn yields between the examined years (Tables 2 and 5). The seed Zn level differed significantly with the different rates of P fertilization. As the P increased, the Zn concentration decreased (Figure 4D). The cultivar was also an important factor for Zn level and yield (Table 2). Among the cultivars, Longya 14 had the highest seed Zn concentration, while Longyaza 1 exhibited the lowest Zn level (Figure 4E). The highest Zn yield was obtained from Longya 14 (Figure 4F).

### 3.6. Manganese Concentration and Yield

The interactions of year, P level, and cultivar significantly affected the Mn concentration and yield (Table 2). In the present study, Mn concentrations varied between 33.0 (Longya 14 at $P_{80}$ in 2019) and 66.3 mg kg$^{-1}$ (Longyaza 1 at $P_0$ in 2018), and Mn yields ranged from 57.4 g ha$^{-1}$ (Longya 14 at $P_0$ in 2019) to 132.4 g ha$^{-1}$ (Longya 14 at $P_{80}$ in 2018) (Table 4). The Mn level and yield were an average of 16.3 mg kg$^{-1}$ and 49.2 g ha$^{-1}$ higher, respectively, in 2018 compared with 2019 (Table 6). The Mn concentration in the flaxseed differed with the different levels of P fertilization (Tables 2 and 6). The seed Mn le vel decreased with an increase in the P level (Figure 4G). Furthermore, the interaction between the year and cultivar influenced the Mn yield. The highest Mn yield was achieved by Longya 14 in 2018 (117.3 g ha$^{-1}$), and the lowest from Longya 14 in 2019 (60.9 g ha$^{-1}$) (Table 7).

**Table 6.** Manganese and copper concentrations and yields for three cultivars and with five phosphorus levels.

| Treatments | Mn [a] Concentration | | Mn Yield | | Cu Concentration | | Cu Yield | |
|---|---|---|---|---|---|---|---|---|
| | 2018 | 2019 | 2018 | 2019 | 2018 | 2019 | 2018 | 2019 |
| | mg kg$^{-1}$ | | g ha$^{-1}$ | | mg kg$^{-1}$ | | g ha$^{-1}$ | |
| $P_0$ | 62.64 a[†] | 45.57 a | 99.98 | 62.58 | 9.01 b | 7.68 b | 14.39 b | 10.57 b |
| $P_{40}$ | 62.42 a | 44.48 a | 104.53 | 67.20 | 10.20 a | 8.83 a | 17.89 a | 13.41 a |
| $P_{80}$ | 60.66 ab | 41.64 b | 106.91 | 65.37 | 10.35 a | 8.90 a | 19.43 a | 14.16 a |
| $P_{120}$ | 56.26 b | 38.79 b | 109.68 | 61.66 | 9.87 ab | 8.78 a | 18.77 a | 14.05 a |
| $P_{160}$ | 50.99 b | 41.05 b | 114.47 | 62.55 | 9.50 ab | 8.13 ab | 17.65 a | 12.52 ab |
| Longyaza 1 | 59.71 | 42.59 b | 102.22 b | 63.55 ab | 9.96 a | 9.16 a | 17.12 | 13.76 a |
| Longya 14 | 58.24 | 36.59 c | 117.32 a | 60.88 a | 9.29 b | 8.14 b | 18.78 | 13.65 a |
| Zhangya 2 | 60.84 | 47.74 a | 101.80 b | 67.18 a | 10.11 a | 8.09 b | 16.99 | 11.42 b |

[a] Mn: manganese; Cu: copper. [†] Means in the same column followed by the same letter do not differ significantly according to the Tukey test ($p = 0.05$). The absence of letters in the same column in dicates no statistically significant differences according to the Tukey test ($p = 0.05$).

**Table 7.** Interactive effect of year and cultivar on the yields of seed, lignan, iron, and manganese, and the concentration and yield of copper.

| Treatment | Seed Yield | Lignan Yield | Iron Yield | Manganese Yield | Copper Concentration | Copper Yield |
|---|---|---|---|---|---|---|
| Lonyaza 1-2018 | 1409.33 c[†] | 16.38 b | 139.44 b | 102.22 b | 9.96 ab | 17.12 a |
| Lonyaza 1-2019 | 1499.33 c | 20.56 a | 179.33 a | 117.32 a | 9.29 ab | 18.78 a |
| Longya 14-2018 | 1668.67 b | 13.69 c | 139.94 b | 101.80 b | 10.11 a | 16.99 a |
| Longya 14-2019 | 1678.03 b | 11.36 d | 106.55 c | 63.55 c | 9.16 b | 13.76 b |
| Zhangya 2-2018 | 1716.37 b | 13.29 c | 125.98 b | 60.88 c | 8.14 c | 13.65 b |
| Zhangya 2-2019 | 2014.16 a | 8.74 e | 106.10 c | 67.18 c | 8.09 c | 11.42 c |

[†] Means in the same column followed by the same letter do not differ significantly according to the Tukey test ($p = 0.05$).

### 3.7. Copper Concentration and Yield

Differences between the two years were observed in terms of the flaxseed Cu concentration and yield (Table 2). The Cu level in 2018, averaging 9.8 mg kg$^{-1}$, was greater than that in 2019 (averaging 8.5 mg kg$^{-1}$). Similarly, the Cu yield was on average 17.6 g ha$^{-1}$ in 2018, which was 4.7 g ha$^{-1}$ greater than that in 2019. (Table 6). In addition, the Cu level and yield were affected by P fertilization, displaying an initial increase followed by a decrease with increasing P fertilization (Figure 4H,I). Moreover, the Cu level and yield were affected by the interaction between the year and cultivar. The highest Cu level and yield were found for Zhangya 2 in 2018 (10.1 mg kg$^{-1}$) and for Longya 14 in 2018 (18.8 g ha$^{-1}$), respectively (Table 7). Furthermore, the interaction between P level and cultivar also affected the Cu yield (Table 2). The lowest value was obtained for Zhangya 2 at $P_0$ (12.2 g ha$^{-1}$), and the highest for Longya 14 at $P_{120}$ (18.4 g ha$^{-1}$) (data not shown).

## 4. Discussion

### 4.1. The Effect of Years

In the current study, the year significantly influenced the seed yield, lignan concentration and yield, dietary fiber concentration and yield, Fe concentration and yield, Zn yield, Mn concentration and yield, and Cu yield. This could be ascribed to differences in and soil nutrients, precipitation, air temperature during the vegetative, flowering, and seed-filling stages of the flax. A previous study indicated that differences in precipitation and air temperature during the vegetative, flowering, and seed-filling stages of flax significantly impact its seed yield, in agreement with the present results [9]. In our study, the year also had a significant effect on the lignan concentration and the dietary fiber concentration. This was in agreement with findings that the year had a significant effect on dietary fiber concentration in Saskatchewan [25]. A previous study found that available Mn in the soil exerted significant positive effects on grain Mn concentration in maize [27]. This was confirmed by the current study. In this case, soil DTPA-Mn at 0–30 cm was greater in 2018 (4.37 mg kg$^{-1}$) than in 2019 (4.01 mg kg$^{-1}$). In addition, studies found that available Cu in soil exerted significant positive effects on grain Cu concentration in maize [27], in line with our results. The lignan, Mn, and Cu yields were greater in 2018 than in 2019, due to the relatively high lignan, Mn, and Cu levels and seed yield in 2018.

### 4.2. Effects of the Phosphorus Rate

In the present study, the seed yield significantly increased with P fertilizer up to 80 kg $P_2O_5$ ha$^{-1}$ and then declined with further increases of the P rate. Excessive P fertilization not only negatively impacted yields in crop production, but also results in environmental, ecological, and human health concerns [28,29]. In the current study, the baseline available P in the soil varied from 8.0 to 8.7 mg kg$^{-1}$ at a depth of 0 to 30 cm, which is considered low [23]. Generally, this study showed that the seed yield significantly increased with application of P fertilizer. This was in agreement with the results of other researchers [8–10], and similar results were found for camellia [28]. The effect of P on the seed yield may be a consequence of increased photosynthesis-related genes at the transcriptional level and enhanced capacity for

light assimilation and efficient photosynthesis [29], as well as improving the source capacity of the flax with the P fertilizer supply.

A study on wheat showed that levels of Fe in grain increased with application of P, compared with plants that did not receive P [13]. This result was agreement with our current findings. However, other studies have reported inconsistent results [12,17,19]. Differences in the Fe concentrations of seed or grain are generally related to differences in genotypes, environment, genotype–environment interactions, and absorption efficiency. In our study, the Zn concentration declined with increased P fertilization, consistent with a previous study on flax [16]. Furthermore, similar findings were achieved in other studies [12–14,17–19]. We also found that the Mn level tended to decline with an increased P application rate, in agreement with findings in canola [12]. Meanwhile, Zn and Mn levels decreased with an increased rate of P application. This may have been due to the seed yield increasing with an increasing P level, resulting in a "dilution effect". In the current study, the Cu level tended to increase first and then decreased. For canola, the P supply had no obvious effects on Cu concentration [12]. In all these cases, the inconsistencies among the different studies can be mainly attributed to the differences in genotypes used, environment, soil nutrient status, and the interactions between these. Meanwhile, the yields of lignan, Fe, and Cu increased with P application, which can mainly be attributed to the fact that the seed yield was enhanced by fertilizer treatments.

### 4.3. *The Effect of Cultivar*

In the current study, the variety of cultivar significantly affected the seed yield. In this respect, our results were in good agreement with previous studies in linseed [30–32]. In a previous investigation conducted in Germany and Spain, the cultivar variety markedly influenced the lignan content [33]. In addition, Zhang et al. [34] found that the lignan level of flaxseed is genetically controlled. Similar results were reported by Garros et al. [35]. Furthermore, researchers reported that the lignan content of sesame was significantly different across elite lines and growing years in Korea [36]. These results also support the findings in the current study. Moreover, we found that the type of cultivar significantly influenced the Zn yield, probably resulting from changes in the seed yield and Zn level.

Flaxseed is an important source of food and nutrition, and its seed yield, lignan yield, and the concentrations and yields of Fe and Cu were positively influenced by P fertilization, while P negatively affected the levels of dietary fiber, Zn, and Mn. In the current study, the highest seed and lignin yields, Fe concentration and yield, and Cu concentration and yield achieved were 1751 kg ha$^{-1}$, 15 g ha$^{-1}$, 84 mg kg$^{-1}$, 145 g ha$^{-1}$, 10 mg kg$^{-1}$ and 17 g ha$^{-1}$, respectively. Furthermore, the seed yield and the levels of lignan, dietary fiber, and Zn, as well as the yields of lignan, dietary fiber, Fe, and Zn, were significantly impacted by differences in cultivar type.

### 5. Conclusions

Appropriate P fertilization is a suitable agronomical and economical approach for improvement of seed productivity and nutrient contents in flaxseed, contributing to food and nutrition security. P application caused a significant increase in seed, lignan, Fe, and Cu yields up to a P rate of 80 kg P$_2$O$_5$ ha$^{-1}$, and Fe and Cu concentrations significantly increased up to P rates of 120 and 40 kg P$_2$O$_5$ ha$^{-1}$, respectively. Concentrations of dietary fiber and Zn markedly declined above 80 and 40 kg P$_2$O$_5$ ha$^{-1}$, respectively. In summary, the application of 80 kg P$_2$O$_5$ ha$^{-1}$ was more favorable for greater yield and nutritional quality of flaxseed. Furthermore, the seed yield and nutritional quality of flaxseed were significantly impacted by the cultivar, and Longya 14 performed well proving good yields and quality.

**Author Contributions:** Conceptualization, methodology, data curation, writing—original draft, writing—review and editing, Y.X. and Y.L.; investigation, Y.Q., L.W., W.Z., W.L., Z.D. and X.W. (Xingzhen Wang); formal analysis, writing—review and editing, J.Z.; project administration, Y.Z., X.W. (Xingrong Wang), Z.C., L.S. and Z.L. All authors have read and agreed to the published version of the manuscript.

**Funding:** This research was funded by the Key Research and Development Projects of Gansu Academy of Agricultural Sciences (2021GAAS20), the National Natural Science Programs of China (31660368; 32060437), the Major Special Projects of Gansu Province (21ZD4NA022-02), and China Agriculture Research System of MOF and MARA (CARS-17-GW-04).

**Data Availability Statement:** The datasets used and/or analyzed during the current study are available from the corresponding author upon reasonable request.

**Acknowledgments:** We gratefully thank the journal's editor and four anonymous reviewers for offering some excellent, constructive suggestions for further improvement our manuscript. We also appreciate the support and help from Junyi Niu and Jianxiang Zhang. We are grateful to Tianqing Yang for technical assistance.

**Conflicts of Interest:** The authors declare no conflict of interest.

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
