# Peer review of "Effects of Phosphorus Supply on Seed Yield and Quality in Flax"

_agronomy, doi:10.3390/agronomy12123225_

Round 1
Reviewer 1 Report
The manuscript “Response of yield, lignan, dietary fibre, iron, zinc, manganese, 2 and copper in flaxseed to phosphorus” is focused on a very informative topic, Overall, I recommend publishing this work in Agronomy (ISSN 2073-4395), as it fits well with the special issue “Effects of Soil Fertility and Plant Growth Promoters on Growth, Yield, and Quality of Crops”. Despite this positive feeling, the authors are requested to consider some minor corrections to improve it before publication.
- Write the practical applications and future research perspectives of this work on one page (before the conclusions section). How can the results of this work be scaled up and applied in other developing countries?
-Introduction should be benefited from the latest literature about Flax (Linum usitatissimum L.).
-Too much phosphorus can be toxic to humans and also to an aquatic environment, which can result in decreased levels of dissolved oxygen. Which evidence do you have to support this claim in your study?
-I strongly encourage authors to update some recent literature citations in the whole article. The following papers can be considered. (https://doi.org/10.1016/j.scitotenv.2022.154043;https://doi.org/10.3390/plants9091173; https://doi.org/10.1016/j.envpol.2021.117916)
-In general, the quality of the figures should be improved since all the pictures are not up to standard, especially Figures 2 and 4.
-A review of the article indicates that the discussion section is poorly written compared to the rest of the article. It would be better if Xie et al. rewrote the section with more care and included more recent studies in support of their study.
-There aren't enough references in the discussion. Please add more references in order to enrich the discussion.
Author Response
The manuscript “Response of yield, lignan, dietary fibre, iron, zinc, manganese, 2 and copper in flaxseed to phosphorus” is focused on a very informative topic, Overall, I recommend publishing this work in Agronomy (ISSN 2073-4395), as it fits well with the special issue “Effects of Soil Fertility and Plant Growth Promoters on Growth, Yield, and Quality of Crops”. Despite this positive feeling, the authors are requested to consider some minor corrections to improve it before publication.
Answer: Firstly, we very appreciate you for recognizing on our manuscript. Secondly, we thank you for offering excellent, constructive suggestions for further improvement the manuscript!
1 - Write the practical applications and future research perspectives of this work on one page (before the conclusions section). How can the results of this work be scaled up and applied in other developing countries?
Answer: Thanks! According to your precious suggestions and combined the other two reviewers’ excellent advices, we rewrote the conclusion, as follows:
“Application appropriate P fertilization was better agronomical and economical approach for improvement in both seed productivity and nutrient content of flaxseed to provide food and nutrition security. Phosphorus application increased significantly the yields of seed, lignan, Fe, and Cu, and the concentrations of Fe and Cu of seed, however, reduced significantly the concentrations of dietary fibre, Zn, and Mn in seed. To sum up, application 80 kg P2O5 ha-1 was more favorable to provide greater yield and nutritional quality of flaxseed. Furthermore, the seed yield and nutritional quality of flaxseed were significantly impacted by cultivar, and Longya 14 performed well with good yields and quality.
To date, biofortification of crops through the P fertilizer application as an effective approach to regulate the contents of mineral elements in crops in many fields especially in developing countries. With the current global increases in environmental and human health and decline in phosphorous ore resource, the P application rate become a focus. There are some concerns regarding the application P rate at different regions, and many works have been done on this aspect. More cost-effective and environmentally-friendly methods of addition P to improve crop productivity and quality for ensuring sustainable food and nutrition security to be developed in the future.”
2 -Introduction should be benefited from the latest literature about Flax (Linum usitatissimum L.).
Answer: Thanks! We did so! The references of [1] and [3] are both the literatures in 2022.
3-Too much phosphorus can be toxic to humans and also to an aquatic environment, which can result in decreased levels of dissolved oxygen. Which evidence do you have to support this claim in your study?
Answer: Thanks! According to your valuable suggestions, we added the sentence “Obviously, excessive P fertilization not only negatively impacted on yields in crop production, but also increased the risk of aquatic ecosystems through eutrophication[29] and heavy metal contamination in rice production [30], which ultimately resulted in a series of concerns for environmental, ecological, and human health [29,31,32]”
4-I strongly encourage authors to update some recent literature citations in the whole article. The following papers can be considered. (https:// doi.org/10.1016/j. scitotenv.2022.154043;https://doi.org/10.3390/plants9091173; https://doi.org/10.1016/j.envpol.2021.117916)
Answer: We thank you very much for offering some excellent, constructive suggestions for further improvement the manuscript! We did so!
5 -In general, the quality of the figures should be improved since all the pictures are not up to standard, especially Figures 2 and 4.
Answer: Thank you very much! We improved them.
6 -A review of the article indicates that the discussion section is poorly written compared to the rest of the article. It would be better if Xie et al. rewrote the section with more care and included more recent studies in support of their study.
-There aren't enough references in the discussion. Please add more references in order to enrich the discussion.
Answer: Thanks! According to your excellent suggestions, we rewrote the section with more care and included more recent studies in support of our study.
Author Response
The study is important and of scientific interest, however, the authors can improve the quality of the title, results, discussion and especially the conclusion that does not respond to the objective and title of the study.
Answer: Firstly, we thank you for recognizing on our manuscript. Secondly, we thank you for offering excellent, constructive suggestions for further improvement the manuscript!
1 Title: The title was very confusing, because the answer is not the variables, but the flaxseed cultivars!
I suggest removing the ratings from the title and changing to "Response of flaxseed cultivars to phosphorus supply on grain yield and quality."
Answer: Thanks! To the best of our knowledge, this word “grain”, it usually refers to cereal crops. Secondly, in the current study, the cultivar affected the seed yield, lignan concentration, dietary fibre level and yield, Fe level, and Zn level and yield, in which the highest values obtained were cultivar Longya 14 (the maximum dietary fibre level observed was Longyaza 1, while there was no significant difference between Longyaza 1 and Longya 14). Moreover, the lowest values were Zhangya 2, exception for Zn level and yield, and there was no significant difference between Longyaza 1 and Zhangya 2. In addition, the interaction between P and cultivar did not affect those traits, except for Cu yield. Hence, be succinct, the title changed to “Response of seed yield and quality of flaxseed to phosphorus supply”. We don't know if you can agree us.
2 Abstract: It was descriptive and covers the entire study.
Answer: Thanks!
3 Material and Methods: It was well described and makes it easy for readers to understand.
Answer: Thanks very much!
4 Results: The results were well presented, however, there was a failure at the time of comparison, at times the authors use the value obtained, at other times they use the percentage of difference. Then take out all the percentages of increase or decrease and accurately describe in the discussion.
Answer: According to suggestion, we took out all the percentages of increase or decrease and described by the value of increase or decrease in the results.
5 Discussion: The discussion was well elaborated, however, there are many comparisons with other species that are copies of the introduction, I don't see the need to repeat this information. Apparently, the authors are justifying the research instead of explaining the results obtained. Be succinct, explain only the results obtained and the reason for the occurrence, the less you compare with results from other research, the greater the relevance of your research.
Answer: Thanks very much! According the suggestions from reviewer, we have revised the discussion.
6 Conclusion: Lines 370-377 - It is not a conclusion, it is an overview of the results obtained, this part of the paragraph should come at the end of the discussion. Be succinct, indicate the ideal dose to provide greater yield, fiber and nutritional quality in the first sentence, thus answering the title and objective of the research.
Answer: Sorry! According to your suggestions and combination the other reviewers advices, we deleted this part. And we rewrote the conclusion.
7 Lines 378-384 - Again it's not discussion, in the last sentence there was the beginning of a conclusion, however, I still couldn't find the ideal dose. If the calculated ideal dose was observed, put it in the conclusion. The conclusion must clearly indicate to readers which dose to use, how it was performed for the ideal cultivar with higher yield and quality.
Answer: Thanks! According to your suggestion, we rewrote the conclusion of manuscript. Now, the conclusion was “Application appropriate P fertilization was better agronomical and economical approach for improvement in both seed productivity and nutrient content of flaxseed to provide food and nutrition security. Phosphorus application increased significantly the yields of seed, lignan, Fe, and Cu, and the concentrations of Fe and Cu of seed, however, reduced significantly the concentrations of dietary fibre, Zn, and Mn in seed. To sum up, application 80 kg P2O5 ha-1 was more favorable to provide greater yield and nutritional quality of flaxseed. Furthermore, the seed yield and nutritional quality of flaxseed were significantly impacted by cultivar, and Longya 14 performed well with good yields and quality.
To date, biofortification of crops through the P fertilizer application as an effective approach to regulate the contents of mineral elements in crops in many fields especially in developing countries. With the current global increases in environmental and human health and decline in phosphorous ore resource, the P application rate become a focus. There are some concerns regarding the application P rate at different regions, and many works have been done on this aspect. More cost-effective and environmentally-friendly methods of addition P to improve crop productivity and quality for ensuring sustainable food and nutrition security to be developed in the future.”
Reviewer 3 Report
Dear Authors,
My comments and suggestions are in the attached file.

Author Response
This article discusses the beneficial effects of P application on the yields of seed, lignin, Fe and Cu as well as enhancing the concentrations of Fe and Cu of flax production. Although the topic is of interest to the Scientific community, before considering it for publication, this paper should be significantly improved, in particular the Discussion chapter. The authors should try to synthesize and highlight the main findings of the study in the Conclusions section.
Answer: Firstly, we thank you for recognizing on our manuscript from your. Secondly, we thank you for offering some excellent, constructive suggestions for further improvement the manuscript!
1 While generally well-written, there are examples of awkward English phrasing and sentence structure that should be corrected before publication. These would be caught by an expert English-language editor upon careful read-through. (ex. Line 55-56 “A study 55 on flaxseed revealed that …” is almost nonsensical as written).
Answer: Thank you for offering some valuable suggestions for further improvement the manuscript. We did our best to improve it! Line 55-56 saw by the following picture:
Here, 55 is line number, please refer to agronomy-2034900-peer-review for details of Line 55-56.
2 The Introduction should be improved. The cited publications are mostly relevant, but a significant number of the references cited are not from the latest 5 years. Authors need to refer to recent articles. The current state of the research field is not clearly highlighted.
Answer: Thanks! According to your suggestions, we deleted the references White and Broadley, 2009 [7], Xie et al., 2014[9], Xie et al., 2016[10], Westcott et al., 2002[29], and Xie et al., 2015 [31], replaced the references [1], [25], [26], [33], and [37] with literatures of Mueed et al., 2022; Recena et al., 2019; Andargie et al., 2021; Li et al., 2022; and Garros et al., 2018, respectively, and added a number of references from the latest 5 years. Moreover, we checked the references cited and changed the Reference number both the text and references throughout the manuscript.
3 Line 59 Year in brackets is not required.
Answer: Thank you very much! We deleted it.
4 Authors did not present any hypothesis at the beginning.
Answer: Thanks! According to your valuable suggestions, we deleted “However, few studies have been performed to examine the impact of P on the levels of mineral micronutrients and yields in flaxseed. In particular, the effects of P on the levels and yields of lignan and dietary fibre in flaxseed remain to be elucidated.”
5 The methodology is good. The conditions of the experiment are described in detail, but some clarifications may be necessary.
Answer: We are very sorry! The study was not the split-plot arrangement. This is a slip in 2.2 Experimental design of the manuscript. It is randomized complete block design. We have amended in 2.2 Experimental design of the current manuscript.
6 Line 82-87 Weather conditions paragraph is rather short. I suggest adding how each year can be characterized in terms of flaxseed production.
Answer: Thanks! We improved “There was appreciable rainfall in July when flax was in the middle seed-filling period in 2018. In 2019, there was more rainfall in August during the later seed-filling periods.” To “Overall, the growing season precipitation, in 2018 there was appreciable rainfall amount resulted from more rainfall frequency and the weather cleared up quickly in July when flax was in the middle seed-filling period, and 2019 there was concentrated rainfall in August during the later seed-filling periods. In the present study, seed yield in 2018 was greater than that of 2019. This due to larger rainfall amount in 2018 in the middle seed-filling period resulted from more rainfall frequency, then the weather cleared up quickly. Under that circumstance, the greater seed yield in 2018 perhaps due to sunshine duration, illumination intensity, humidity, soil moisture, soil microorganisms, etc. were more favorable seed-filling. However, in 2019, in the condition of concentrated rainfall, over rainfall caused serious conflicts between water supply and crop demands, and led to lodging and disease.”
7 Line 88-93 and Line 114-123 The soil characteristics are described in two separate paragraphs in the Material and Method sections. I propose to summarise these in a sub-chapter. I also propose to summarise the chemical characteristics of soil in a table, either by extending Table 1.
Answer: Thanks! According to your suggestion, 2.1 Experimental Site and 2.3 Sampling and measurement summarized in a subchapter 2.1 Experimental Site. And we extended Table 1.
8 Line 98 Please correct the title of Table 1
Answer: Thanks! The title of Table 1 was improved “Table 1. General soil properties and DTPA extractable mineral elements before sowing in 2018 and 2019.”
9 Line 101-102 Authors need to explain why they have selected these flaxseed varieties.
Answer: Thanks! To address this issue, we added the sentence “The cultivars were mainly planted of local agriculture department and farms.” In 2.2 Experimental design.
10 Line 111-112 How was the timing of irrigation and the amount of water used determined? In June 2018, rainfall was above average, as shown in Figure 1. What irrigation method was used?
Answer: The timing of irrigation was at key growth stages of crop demands, and the amount of water was appropriate for plants of flax demand. Over irrigation causes serious conflicts between water supply and crop demands. The timing of irrigation and the amount of water were determined based on local long-term agriculture department and farms management practices. Traditional irrigation was used with pipe. Pipes with diameter of 13cm were used for irrigation, and a water meter installed at discharging end of the pipe was to measure and record the irrigation amount.
11 Line 122-123 What was the date of harvesting?
Answer: Thanks! In 2.2 Experimental design, we added “2 August 2018 and 3 August 2019”
12 Line 159 Please correct the sentence “The lignan concentration….”
Answer: Thanks! The sentence “The lignan concentration increased by an average of 28% in 2018 than that of 2019.” have been improved to “The lignan concentration was 2.0 g kg-1 greater in 2018 than that of 2019.”
13 Line 251 In Figure 4, the diagrams are too small, making it difficult to read the axis labels.
Answer: We felt very sorry! Now, we enlarged the diagrams in Figure 4.
14 Line 315-317 Please clarify this conclusion: “The 315 lignan, Mn, and Cu yields…”
Answer: Thank you suggestion for improving our manuscript! Details of line 315-317 saw by the following picture:
Here, 315 is line number, please refer to agronomy-2034900-peer-review for details.
15 The authors lack to highlight what is new. The findings of the paper are not surprising, generally in agreement with previous studies on similar crops
Answer: Thanks! According to your valuable suggestions, we rewrote some contents.
16 I suggest to further develop the conclusions. Synthetize in 3-5 bullet the key results of the study, evidences and recommendation. Add a practical implications statement.
Answer: Thank you for offering precious suggestion and hints! Combination the suggestions from the other two reviewers. We rewrote the conclusion of manuscript. Now, the content of conclusion is as follows:
Application appropriate P fertilization was better agronomical and economical approach for improvement in both seed productivity and nutrient content of flaxseed to provide food and nutrition security. Phosphorus application increased significantly the yields of seed, lignan, Fe, and Cu, and the concentrations of Fe and Cu of seed, however, reduced significantly the concentrations of dietary fibre, Zn, and Mn in seed. To sum up, application 80 kg P2O5 ha-1 was more favorable to provide greater yield and nutritional quality of flaxseed. Furthermore, the seed yield and nutritional quality of flaxseed were significantly impacted by cultivar, and Longya 14 performed well with good yields and quality.
To date, biofortification of crops through the P fertilizer application as an effective approach to regulate the contents of mineral elements in crops in many fields especially in developing countries. With the current global increases in environmental and human health and decline in phosphorous ore resource, the P application rate become a focus. There are some concerns regarding the application P rate at different regions, and many works have been done on this aspect. More cost-effective and environmentally-friendly methods of addition P to improve crop productivity and quality for ensuring sustainable food and nutrition security to be developed in the future.
Reviewer 4 Report
Line 20-"trails" should be trials
Line 21-remove "of" and put "2018 and 2019" in brackets
Line 22-insert the word "application" before "...rates of phosphorus.."
Line 24-insert word "fertilization" after P, after ",...seed" insert "and contents of"
Line 25-remove "in the presence of P fertilization" it is redundant.
Line 26-remove "an"
Line 29-following "the P level..." add "application"
Line 31-replace "one of" with "an", change strategies to strategy
Line 33-change to "Cu in flax production."
etc
Author Response
1 Line 20-"trails" should be trials
Answer: We must thank you for being so kind to us! We did so!
2 Line 21-remove "of" and put "2018 and 2019" in brackets
Answer: Thanks! We did so!
3 Line 22-insert the word "application" before "...rates of phosphorus.."
Answer: Thanks! We did so!
4 Line 24-insert word "fertilization" after P, after ",...seed" insert "and contents of"
Answer: Thanks! We did so!
5 Line 25-remove "in the presence of P fertilization" it is redundant.
Answer: Thanks! We did so!
6 Line 26-remove "an"
Answer: Thanks! We did so!
7 Line 29-following "the P level..." add "application"
Answer: Thanks! We did so!
8 Line 31-replace "one of" with "an", change strategies to strategy
Answer: Thanks! We did so!
9 Line 33-change to "Cu in flax production."
Answer: Thank you very much for offering precious suggestions for further improvement the manuscript! We did so!
Round 2
Reviewer 3 Report
I accept the corrected manuscript.